# Analysis of Pacing Behaviors on Mass Start Speed Skating

**DOI:** 10.3390/ijerph191710830

**Published:** 2022-08-30

**Authors:** Qian Peng, Feng Li, Hui Liu, Miguel-Angel Gomez

**Affiliations:** 1School of Competitive Sports, Beijing Sport University, Beijing 100084, China; 2China Basketball College, Beijing Sport University, Beijing 100084, China; 3China Institute of Sports and Health, Beijing Sport University, Beijing 100084, China; 4Faculty of Physical Activities and Sport Sciences, Polytechnic University of Madrid, 28040 Madrid, Spain

**Keywords:** performance analysis, pacing strategy, winning strategy, Winter Olympic Games

## Abstract

The mass start speed skating (MSSS) is a new event for the 2018 PyeongChang Winter Olympic Games. Considering that the event rankings were based on points gained on laps, it is worthwhile to investigate the pacing behaviors on each lap that directly influence the skater’s ranking at the end of the race. To the best of our knowledge, this is the first study investigating the pacing behavior on the MSSS. The aim of this study was to analyze the pacing behaviors and performance on MSSS regarding skaters’ level (SL), competition stage (semi-final/final) (CS), and gender (G). All the male and female races in the World Cup and World Championships were analyzed during the 2018–2019 and 2019–2020 seasons. As a result, a total of 601 skaters (male = 350 and female = 251) from 36 games (male = 21 and female = 15) were observed. The one-way ANOVA for repeated measures was used to compare skaters’ pacing behavior on each lap, and the three-way ANOVA for repeated measures was used to identify the influence of SL, CS, and G on skaters’ pacing behaviors and total time spent. In general, the results showed that the pacing behaviors from fast to slow were group one (laps 4, 8, 12, 15, 16), group two (laps 5, 9, 13, 14), group three (laps 3, 6, 7, 10, 11), and group four (laps 1 and 2) (*p* ≤ 0.001 for all groups). For CS, the total time spent in the final was less than the semi-final (*p* ≤ 0.001). For SL, top-level skaters spent less total time than the middle-level and low-level skaters (*p* ≤ 0.002), while there was no significant difference between the middle and low levels (*p* = 0.214). For G, the male skaters spent less total time than females on all laps (*p* ≤ 0.048). Current findings could help coaching staff to better understand the pacing behaviors regarding SL, CS, and G. In particular, the identified performance trends may allow controlling for pacing strategy and decision making before and during the race.

## 1. Introduction

The mass start speed skating (MSSS) is a newly added event for the 2018 PyeongChang Winter Olympic Games. The distance of the MSSS is 6400 m, which is a long-distance speed skating race and requires appropriate pacing strategy for better ranking [1,2]. Unlike other types of speed skating race, the MSSS consists of 16 laps, and 4 points laps are set at the end of the 4th, 8th, 12th, and 16th laps. According to the revised rules in 2018, the top 3 skaters on the 4th, 8th, and 12th laps can be awarded with 3, 2, and 1 point, respectively, whereas the top 6 skaters on the 16th lap (last lap) can be awarded with 60, 40, 20, 10, 6, and 3 points, respectively. Consequently, the ranking of the race depends on the number of points the skaters earned. Favorable pacing behavior plays an important role for the long-distance races in achieving good performances [3,4,5,6,7]. For example, in order to achieve the best performance in the final race, skaters tend to save energy at non-full speed in the semi-final as long as they can be qualified for the final, which can then be beneficial for the final outcome [8,9,10]. In addition, MSSS is a same-track racing that accommodates up to 24 skaters that simultaneously start from the same starting line. Thus, it is important for coaches to understand the pacing behavior on each lap in terms of the world-level competitions as well as to understand how to adopt pacing strategies in the semi-final and the final, which directly determinate the final ranking of the race.

In the past, there have been several studies investigating the winning strategy of MSSS [1,11]. However, those studies only examined either one event or one race accompanying fewer games. Importantly, the MSSS rules have been revised since the 2018–2019 season, whereas the samples of previous studies were observed in the 2017–2018 season. Thus, it can no longer provide references for coaches and researchers. On the other hand, several studies have observed the pacing behaviors on speed skating while competing on multiple tracks [12,13,14,15,16], while all skaters in the MSSS compete on the same track. Furthermore, some studies have investigated the pacing behaviors and winning strategies on short track speed skating that compete in the same track [9,10,15,17], while the distance of the race is shorter (the longest individual event is 1500 m). Therefore, the pacing behavior in terms of the MSSS is still unknown. Understanding the pacing behavior of SL, CS, and G could help coaching staff have a better understanding of MSSS characteristics, design appropriate training sessions, and implement a pacing strategy during the race. As a result, it is vital to investigate the differences in pacing behaviors according to SL, CS, and G.

To the best of our knowledge, this is the first study investigating the pacing behavior on the MSSS. Therefore, the current study aimed to identify the impact of SL, CS, and G on the pacing behavior and the performance of the MSSS. It was hypothesized that pacing behaviors from fast to slow would be group one (laps 4, 8, 12, 15, and 16), group two (laps 5, 9, and 13), group three (laps 2, 3, 7, 11, and 14), and group four (lap 1, 6, and 10). In addition, it was hypothesized that SL, CS, and G would affect skaters’ performances with faster behaviors in top-level skaters compared to middle and low-level, during the final compared to the semi-final, and for males compared to females. 

## 2. Materials and Methods

### 2.1. Samples

In total, data from 8 World Speed Skating Cups and 2 World Single Distance Championships in Men’s and Women’s MSSS during the 2018–2019 and 2019–2020 seasons were gathered. The sample comprised 36 races after excluding 2 races (36 skaters) with incomplete data. Altogether, after excluding 41 skaters who abstained, fouled, or did not complete the race, 601 skaters (male = 350 and female = 251) were collected. Generally, 17 races in the semi-final (male = 11 and female = 6) and 19 races in the final (male = 10 and female = 9) were analyzed. The samples are defined in Table 1. The research followed the European Data Protection Law and kept the anonymity of athletes.

### 2.2. Instruments

This study was a retrospective analysis of publicly available data from the official website of the International Skating Union (ISU) [18] (https://app.isuresults.eu/events; accessed on 8 December 2020). All events, including the Olympic Games, World Championships, European Championships, ISU World Cup, and Four Continents competitions, are described in detail on the website. It also includes skater profiles, race results, and skaters’ performances.

### 2.3. Procedure

Data obtained using the aforementioned methodology across different contexts have exhibited good reliability and validity [8,9,10,17]. The data examined included the ranking, total time spent, lap time spent (1–16 laps), points skaters earned on points laps (4th, 8th, 12th, and 16th laps), and the total points skaters earned in the race. In order to identify the skaters’ performance based on their different levels, skaters were categorized as top-level, middle-level, and low-level according to the points players earned in each race. The top 3 skaters in the race were classified as top-level (male = 63 and female = 45), the skaters who earned points but were not in the top 3 were classified as middle-level (male = 143 and female = 113), and the skaters who did not earn any points were classified as low-level (male = 144 and female = 93). The categorization was in accordance with previous studies related to short track speed skating [8,9,10,17]. 

### 2.4. Statistical Analysis

First, data normality assumptions were checked using the Kolmogorov–Smirnov (KS) test. Secondly, the one-way ANOVA for repeated measures was run to test the speed differences on each lap, and the independent variable was lap number (1–16 laps), whereas the dependent variable was the lap time spent. In order to check the pairwise comparisons, the LSD post hoc test was applied. Thirdly, a three-way ANOVA for repeated measures was used to identify the influence of CS, SL, and G on the total time spent and the lap time spent. The independent variables were CS, SL, and G, whereas the dependent variables were the total time spent and the lap time spent. The effect size was determined using the partial eta squared (η^2^p), which is suggested by Lakens [19] and Richardson [20]. Partial eta squared values of 0.01, 0.06, and 0.14 indicated a small, medium, and large effect of measurements, respectively [21]. All statistical analysis were run using the statistical software IBM SPSS statistics version 21.0 (IBM. Corp., Armonk, NY, USA). The significant level was set at *p* < 0.05. We used the mark “≤” to summarize the statistically significant differences in the results due to the large number of *p* values obtained after comparing each of the 16 laps (e.g., the lap 1 compare with lap 2, 3 until lap 16; the lap 2 compare with 3, 4 until 16, and so on). Likewise, we used the mark “≥” to summarize the statistically insignificant differences in the results.

## 3. Results

### 3.1. The Pacing Behavior

The ANOVA for repeated measures (see Table 2 and Figure 1) showed that there was a main effect on lap time spent (*p* ≤ 0.001). The results showed that the pacing behavior from fast to slow were group one—laps 4, 8, 12, 15, and 16, group two—laps 5, 9, 13, 14, group three—laps 3, 6, 7, 10, 11, and group four—laps 1 and 2 (*p* ≤ 0.001 for all groups).

### 3.2. Influence of CS, SL, and G on Total Time Spent

The three-way ANOVA analysis revealed that: (i) there was a main effect between total time spent and CS, showing that the speed in the final was significantly faster than the semi-final (*p* < 0.001); (ii) there was a main effect between total time spent and SL (*p* < 0.001), showing that top-level skaters were significantly faster than middle-level and low-level skaters (*p* ≤ 0.002), while there was no significant difference between the middle-level and low-level skaters (*p* = 0.214); and (iii) there was a main effect between total time spent and G, showing that male skater speed was significantly faster than female (*p* < 0.001).

### 3.3. Influence of CS, SL, and G on Pacing Behavior

As shown in Table 2 and Figure 2, the three-way ANOVA analysis demonstrated that there was a main effect for G on each lap time spent, showing that male skaters were significantly faster than female (*p* ≤ 0.048).

Additionally, there was a main effect for CS on laps 1, 4, 5, 6, 7, 9, 10, 11, 12, 13, 14, and 15 (*p* ≤ 0.038) (Figure 3). The speed on lap 1 was significantly faster in the semi-final than the final (*p* = 0.038), while the speed on other laps above was significantly faster in the final than the semi-final (*p* ≤ 0.02).

Furthermore, there was a main effect for SL on laps 4, 8, 9, 12, 13, 15, and 16 (*p* ≤ 0.014) (Figure 4): (i) The speed on laps 4, 8, 12, and 15 was significantly faster in the top level than the middle and low levels (*p* = 0.006); (ii) the speed on lap 9 was significantly faster in the top and low levels than the middle level (*p* = 0.024), while there were no significant differences between the top and low levels (*p* = 0.923); (iii) the speed on lap 13 was significantly faster in the low level than the middle level (*p* = 0.004), while there were no significant differences among the top, middle, and low levels (*p* = 0.123); and (iv) the speed on lap 16 was significantly faster in the top level than the middle and low levels (*p* ≤ 0.001), and the middle level was significantly faster than the low level (*p* = 0.006).

On the other hand, an interaction effect was found for G and CS on laps 3, 4, 9, 11, 13, 14, and 15 (*p* ≤ 0.024) (Figure 5). For male skaters, the speed on all laps was significantly faster in the final than the semi-final (*p* < 0.001), while, for female skaters, the speed on laps 9, 11, 13, 14, and 15 was significantly faster in the final than the semi-final (*p* < 0.001).

Similarly, an interaction effect was found for CS, SL, and G on lap 15 (*p* = 0.036) (Figure 6). In terms of CS, for male skaters, there were no significant speed differences in the semi-final among the top, middle, and low levels (*p* ≥ 0.210). However, the male top-level skaters were significantly faster in the final than the low-level skaters (*p* = 0.021). For female skaters, the speeds of the top and middle levels in the semi-final were significantly faster than the low level (*p* ≤ 0.018), while the speed of the female top-level skaters in the final was significantly faster than the middle and low levels (*p* ≤ 0.038). In terms of SL, for male skaters, there was no significant difference for the top and low levels between the semi-final and the final (*p* ≥ 0.070). Conversely, the speed of the men’s middle level in the final was faster than in the semi-final (*p* = 0.014). For female skaters, the speed of the top, middle, and low levels were all significantly faster in the final than in the semi-final (*p* ≤ 0.013). In terms of G, the male skaters’ speed was significantly faster than the females’ in the semi-final (*p* ≤ 0.008). Additionally, the male skaters’ speed was significantly faster than the females’ either for the middle level or the low level in the final (*p* ≤ 0.009), while there was no significant difference on lap 15 between the male top level and female top level in the final (*p* = 0.070).

## 4. Discussion

The MSSS rules were revised after the PyeongChang Winter Olympics in 2018. To the best of our knowledge, this is the first study investigating the pacing behavior and performance in the MSSS. The aim of this study was to analyze the pacing behaviors and performance on MSSS regarding SL, CS, and G. From a general point of view, the results of this study indicated that the pacing behavior from fast to slow were group one (laps 4, 8, 12, 15, 16), group two (laps 5, 9, 13, 14), group three (laps 3, 6, 7, 10, 11), and group four (laps 1 and 2). Additionally, the pacing behavior in the final was significantly faster than the semi-final; the top-level skater was faster than the middle- and low-level skaters, while there was no significant difference between the middle and low levels. Moreover, male skaters’ speed was significantly faster than females’ on all laps. 

It was hypothesized that pacing behaviors from fast to slow would be group one (laps 4, 8, 12, 15, and 16), group two (laps 5, 9, and 13), group three (laps 2, 3, 7, 11, and 14), and group four (lap 1, 6, and 10). In addition, it was hypothesized that SL, CS, and G would affect skaters’ performances with faster behaviors in top-level skaters compared to middle- and low-level skaters, during the final compared to the semi-final, and for males compared to females. The findings are partly in line with our previously formulated hypotheses.

### 4.1. The Pacing Behavior

The pacing behaviors from fast to slow were group one (laps 4, 8, 12, 15, 16), group two (laps 5, 9, 13, 14), group three (laps 3, 6, 7, 10, 11), and group four (laps 1 and 2). For group one, it is well understood that skaters increased their speed in order to obtain points on the points laps (i.e., laps 4, 8, 12, and 16). Additionally, the notable speed on the 15th lap may be explained by the fact that skaters require an advantageous position when they enter the 16th lap to achieve a better ranking. The 4th lap was faster than laps 8, 12, and 16, which may be related to the fact that the skaters have sufficient energy in the early stage and tend to obtain an advantageous position. For group two, a very possible explanation is that after sprinting with high speed to obtain points on laps 4, 8, 12, the skaters are able to maintain higher speed on laps 5, 9, and 13 due to the inertial factor (skaters do not need to sprint with high speed on laps 5, 9, and 13, but their speed remains very high since they can obtain extra speed from laps 4, 8, and 12). Although the 14th lap is not one of the points laps, it is relatively closer to the last lap, so skaters tend to speed up on this lap. For group three, the speed on laps 3, 7, and 11 was relatively slower, demonstrating that skaters tend to save energy to prepare for the point laps (laps 4, 8, and 12); the lower speed on laps 6 and 10 may be due to the fact that the inertial factor (extra speed obtained from laps 4 and 8) was lower compared to laps 5 and 9 after the skaters crossed the points laps (4th and 8th). For group four, the speed on laps 1 and 2 was lowest compared to the other laps, which may be due to two aspects. First, many studies have reported that the athletes’ speed is lower on the first several laps in long-distance races to save energy [9,17,22,23,24]. Secondly, the rules of the MSSS stipulate that skaters are not allowed to surpass others on the first lap [2]. Consequently, skaters’ speed on the first and second laps was lower compared to other laps.

### 4.2. Influence of CS, SL, and G on Total Time Spent

The results indicated that the total time spent for male skaters was significantly lower than female, which is consistent with previous studies showing that male athletes were faster than females in similar race events [8,9,10,17]. Furthermore, the total time spent in the final was significantly lower than the semi-final, which may be explained by the fact that the definition of different levels of skaters between semi-final and final was different (i.e., the low-level skaters in the semi-final may not qualify for the final, while the low-level skaters in the final may be the middle-level skaters in the semi-final). Moreover, the total time spent for top-level skaters was lower than middle- and low-level skaters, which is because the top-level skaters were categorized as the three skaters obtaining the most points in the race (i.e., the three fastest skaters to finish the race). Thus, the top-level skaters in this study spent less time than other skaters in the race. However, there were no significant differences between middle- and low-level skaters. This finding may be explained by the fact that the total time spent does not reflect the order of the rankings according to the rules of the MSSS except for the top three skaters. For example, the middle-level skaters’ ranking is better than the low-level, but the middle-level skaters may spend more total time than low-level skaters.

### 4.3. Influence of CS, SL, and G on Pacing Behavior

For CS, there were 11 laps that were significantly faster in the final than the semi-final. A possible explanation for this result might be related to the SL. As explained above, the speed on the final was expected to be higher since the definitions of the different levels of skaters between the semi-final and final were different (e.g., the low-level skaters in the semi-final may not qualify for the final, while the top-level skaters in the semi-final may be defined as middle-level in the final). Thus, the pacing behavior in the final was faster than the semi-final in most laps. Furthermore, there was no significant difference on the 8th lap between the semi-final and the final, implying that skaters tend to adjust their speed after half of the race and adopt a different pacing strategy in the next half of the race. Additionally, there was no significant difference on the 16th lap, which is due to the fact that the 16th lap has the highest number of points, and all skaters tend to sprint at full speed on this lap in both the semi-final and the final. For SL, the speeds on 4th, 8th, and 12th laps were significantly faster for the top and middle level skaters than low-level skaters, showing that the top and middle levels have a stronger ability to obtain points than low-level skaters. However, the speed on the 15th and 16th laps was significantly faster in top-level skaters than both middle- and low-level skaters, implying that the sprint ability of the top-level skaters on the last two laps was stronger than that of the middle- and low-level skaters. Additionally, the speeds on the 9th and 13th laps were significantly faster in the top- and low-level skaters than the middle-level, which may be explained by the fact that although 9th and 13th were under the buffer period after point laps (8th and 12th laps), top-level skaters tend to maintain high speeds after half of the race for a better result; the low-level skaters’ speeds increased since they did not obtain points on the first three points laps. Consequently, the top- and low-level skaters’ speeds were higher on the 9th and 13th laps than the middle-level skaters. For G, the male skaters’ speed on each lap was significant faster than the female skaters’, which is due to the fact that male athletes have better physical and physiological abilities than female athletes [8,9,10,17].

The results showed that the male skaters’ speed in the final was significantly faster than the semi-final on all laps above, which may be explained by the different definitions of the different levels of the skaters between the semi-final and final (i.e., the low-level skaters in the semi-final may not qualify for the final, while the low-level skaters in the final may be the middle-level skaters in the semi-final). The female skaters’ speed did not differ between the semi-final and the final on lap 4, indicating that female skaters tend to pay more attention to the first point lap (lap 4) in the semi-final than in the final. Thus, they increased their speed on lap 3 in order to earn points on lap 4, resulting in no significant difference on lap 4 between the semi-final and the final.

There was an interaction for G, CS, and SL on the 15th lap. In terms of CS, for male skaters, the further statistical analysis showed that there was no significant difference in the semi-final among male top-, middle-, and low-level skaters, which may be explained by the fact that the 15th lap plays an important role in determining the result of the race, which means that all levels of skaters tend to sprint at full speed to achieve a better ranking. Furthermore, the male top-level skaters were significantly faster than the low-level skaters in the final, while there were no significant differences between the male top- and middle-levels and between male middle- and low-levels. This finding is explained by the fact that the gap between top-level and low-level skaters is larger than the gap between middle-level and low-level skaters in the final. For female skaters, the top- and middle-level skaters in the semi-final were significantly faster than low-level skaters, while there was no significant difference between the top- and middle-level skaters. In contrast, the top-level skaters in the final were significantly faster than middle- and low-level skaters, while there were no significant differences between middle- and low-level skaters. These findings are likely to be explained by the fact that female skaters’ competition was fiercer in the final than the semi-final, meaning that the female top-level skaters in the semi-final tend to save energy as long as they can qualify for the final. Conversely, they tend to speed up on the 15th lap to achieve a better ranking in the final. In terms of SL, for male skaters, there was no significant difference between the semi-final and the final regarding the top-level and low-level, which may be explained by the same fact that skaters speed up on the 15th lap since this lap is close to the finish line. In contrast, male middle-level skaters in the final were significantly faster than in the semi-final. A possible explanation may be that the definition of the middle level is different between the semi-final and final (i.e., middle-level skaters who hardly make it from the semi-final to the final may be considered low-level in the final, and top-level skaters in the semi-final may be middle-level in the final). For female skaters, the top-, middle-, and low-level skaters were all significantly faster in the final than the semi-final, which may be explained by the same fact that the definition of the SL was different between the semi-final and final. In terms of G, the male skaters’ speed was significantly faster than the females’ in the semi-final regardless the SL, which may be due to the differences of physical fitness between males and females [8,9,10,17]. However, there was no significant difference on 15th lap between top-level male and female skaters in the final, implying that only when female skaters reach speeds similar to males on the 15th lap can they achieve better results in the final. Thus, this result may provide a reference for coaches to design an appropriate training regimen to improve female skaters’ sprint ability, especially on the 15th and 16th laps.

There are limitations in the current study that need to be considered in the future. The MSSS was introduced to the winter Olympics for the first time in 2018, so there have been fewer world competitions. Accordingly, future investigations of large datasets are encouraged to further study the pacing behavior in the MSSS. Additionally, future studies are recommended that examine the pacing behavior in events of different levels such as continental and national competitions.

## 5. Conclusions

To the best of our knowledge, this is the first study providing objective, scientific analysis regarding pacing behavior in the MSSS. The following conclusions can be drawn from the present study:

For pacing behavior, the points laps (4th, 8th, 12th, and 16th) have primary influence on pacing behavior in MSSS. To be specific, first, the skaters’ speed was the slowest at the beginning of the competition (1st and 2nd laps). Secondly, the speed was fastest on the 4th, 8th, 12th, 15th, and 16th laps. Thirdly, there were several buffer laps before and after the first 3 points laps. Lastly, the speed on the 15th lap significantly increased. 

For the influence of SL, CS, and G on total time spent, the total time spent of the top-level skaters was significantly lower than the middle-level and low-level skaters, while there was no significant difference between the middle and low levels. Additionally, the final was significantly lower than the semi-final. Furthermore, the total time spent for male skaters was significantly lower than for female skaters.

For the influence of SL, CS, and G on pacing behaviors, the skaters’ speed in the final was significantly faster than the semi-final, and the skaters’ speed at the beginning was similar between the semi-final and the final. Furthermore, the top-level skaters had better sprint ability and better pacing strategy than other levels of skaters. Moreover, the male skaters’ speed was significantly faster than the female skaters’ during most of the laps. Importantly, the current study confirmed that the 15th lap plays an important role in determining the final ranking. 

In general, this study suggests that in order to achieve a better result, skaters’ ideal pacing behavior may involve the slowest speed on the 1st and 2nd laps and fastest speed on 4th, 8th, 12th, 15th, and 16th laps. Furthermore, we propose that skaters are better able to reach the leading position on the 15th lap to increase the final ranking. Moreover, skaters generally need to have faster speeds in the final than the semi-final in order to achieve better rankings. This research could help coaching staff to better understand the top skaters’ pacing behaviors, further providing references for promoting pacing strategy and decision making before and during the race.

## Figures and Tables

**Figure 1 ijerph-19-10830-f001:**
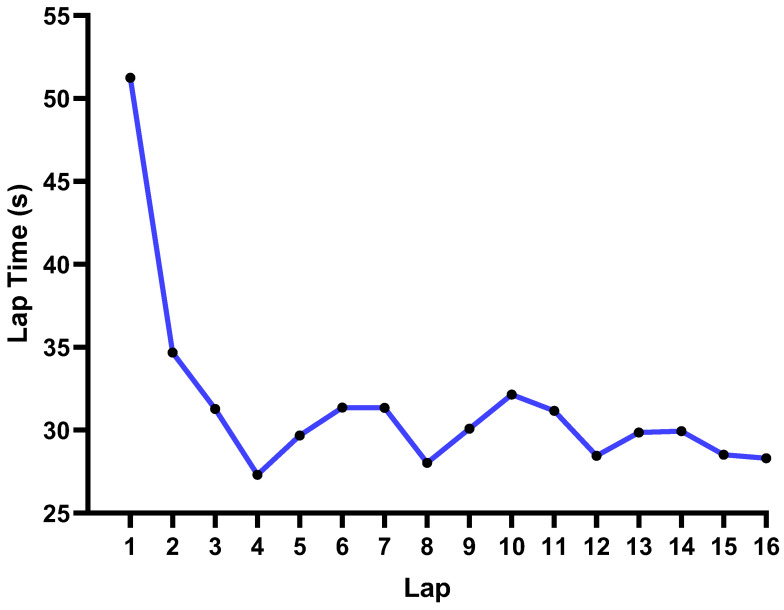
Lap time spent on each lap.

**Figure 2 ijerph-19-10830-f002:**
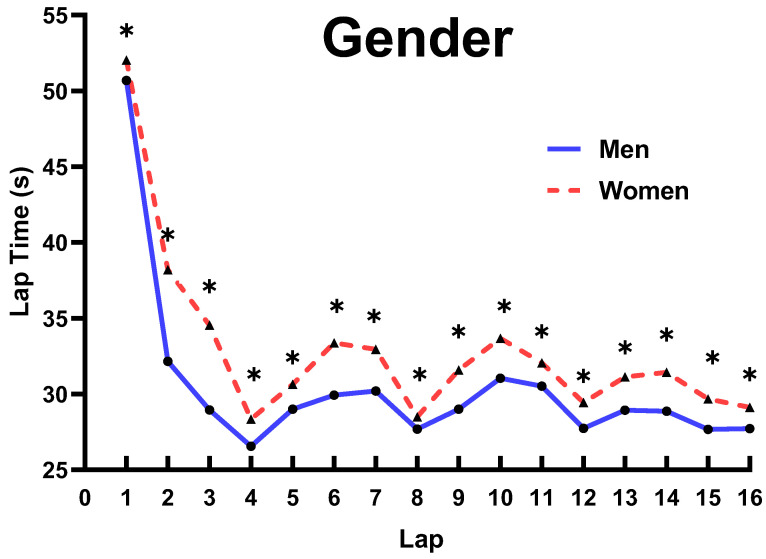
Comparison on lap time spent for G. Note: * marked values were significant when *p* ≤ 0.05.

**Figure 3 ijerph-19-10830-f003:**
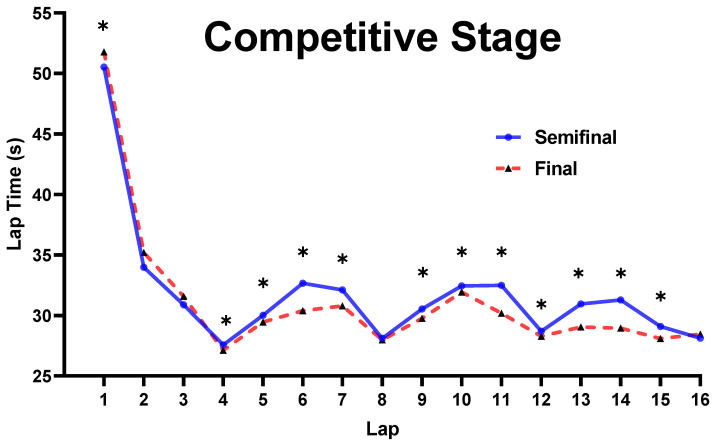
Comparison of lap time spent for CS. Note: * marked values were significant when *p* ≤ 0.05.

**Figure 4 ijerph-19-10830-f004:**
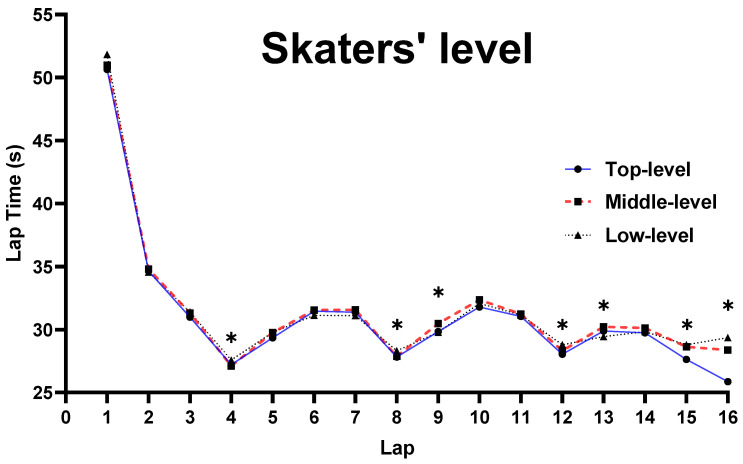
Comparison of lap time spent for SL. Note: * marked values were significant when *p* ≤ 0.05.

**Figure 5 ijerph-19-10830-f005:**
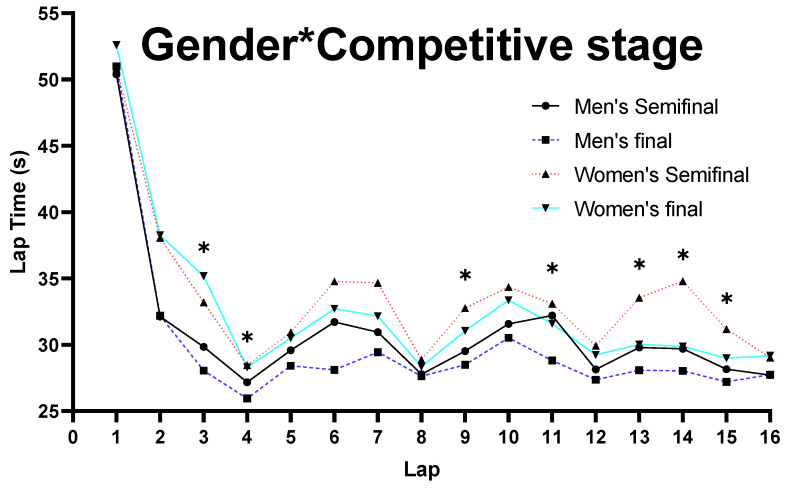
Comparison of lap time spent between G and CS. Note: * marked values were significant when *p* ≤ 0.05.

**Figure 6 ijerph-19-10830-f006:**
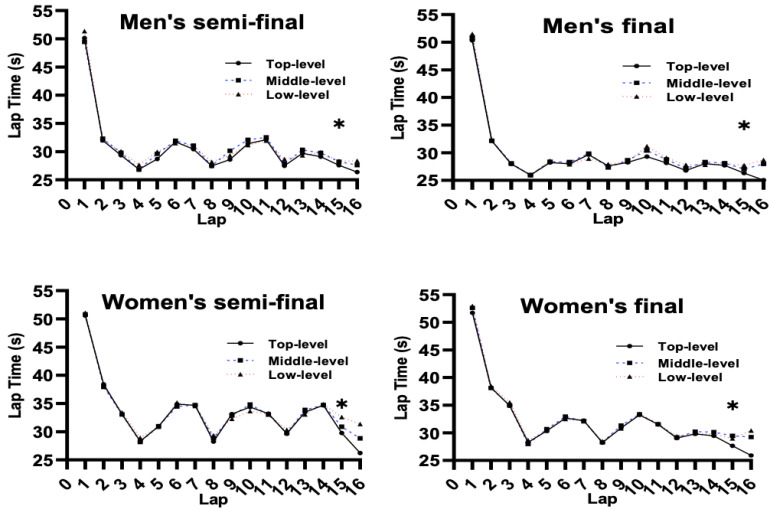
Comparison of lap time spent among G, CS, and SL. Note: * marked values were significant when *p* ≤ 0.05.

**Table 1 ijerph-19-10830-t001:** Sample from the World Cup and World Championships during the 2018–2019 and 2019–2020 seasons.

CS	G (Number of Races)	SL	Number of Skaters
Final	Male (n = 10)	Top	30
		Middle	74
		Low	69
	Female (n = 9)	Top	27
		Middle	76
		Low	68
Semi-final	Male (n = 11)	Top	33
		Middle	69
		Low	75
	Female (n = 6)	Top	18
		Middle	37
		Low	25

Note: G-gender; CS-competitive stage (semi-final/final); SL-skaters’ level.

**Table 2 ijerph-19-10830-t002:** Three-ways ANOVA for each lap time and total time spent.

Laps		G	CS	SL	G × CS	G × SL	CS × SL	G × CS × SL
L1	F	3.909	4.324	1.567	1.050	0.408	0.260	0.176
	*p*	0.048 *	0.038 *	0.210	0.306	0.665	0.771	0.839
	η^2^p	0.007	0.007	0.005	0.002	0.001	0.001	0.001
L2	F	203.658	0.059	0.004	0.004	0.101	0.083	0.054
	*p*	0.000 *	0.808	0.996	0.951	0.904	0.920	0.947
	η^2^p	0.257	0.000	0.000	0.000	0.000	0.000	0.000
L3	F	306.538	0.110	0.659	36.570	0.166	0.023	0.193
	*p*	0.000 *	0.741	0.518	0.000 *	0.847	0.977	0.825
	η^2^p	0.342	0.000	0.002	0.058	0.001	0.000	0.001
L4	F	246.803	30.717	10.419	16.816	0.769	2.072	1.403
	*p*	0.000 *	0.000 *	0.000 *	0.000 *	0.464	0.127	0.247
	η^2^p	0.295	0.050	0.034	0.028	0.003	0.007	0.005
L5	F	62.360	10.983	0.975	1.606	0.992	0.234	0.464
	*p*	0.000 *	0.001 *	0.378	0.206	0.372	0.792	0.629
	η^2^p	0.096	0.018	0.003	0.003	0.003	0.001	0.002
L6	F	87.308	48.875	0.065	3.335	0.194	0.219	0.225
	*p*	0.000 *	0.000 *	0.937	0.068	0.823	0.803	0.799
	η^2^p	0.129	0.077	0.000	0.006	0.001	0.001	0.001
L7	F	78.010	28.018	0.281	2.190	0.134	0.271	0.330
	*p*	0.000 *	0.000 *	0.755	0.139	0.874	0.763	0.719
	η^2^p	0.117	0.045	0.001	0.004	0.000	0.001	0.001
L8	F	31.811	3.145	5.931	1.707	0.806	1.118	0.139
	*p*	0.000 *	0.077	0.003 *	0.192	0.447	0.328	0.870
	η^2^p	0.051	0.005	0.020	0.003	0.003	0.004	0.000
L9	F	260.855	53.330	4.988	5.096	2.030	0.369	1.844
	*p*	0.000 *	0.000 *	0.007 *	0.024 *	0.132	0.692	0.159
	η^2^p	0.307	0.083	0.017	0.009	0.007	0.001	0.006
L10	F	68.390	9.984	0.911	0.232	0.681	2.433	0.220
	*p*	0.000 *	0.002 *	0.403 *	0.630	0.507	0.089	0.802
	η^2^p	0.104	0.017	0.003	0.000	0.002	0.008	0.001
L11	F	46.039	81.354	0.207	12.360	0.353	0.459	0.288
	*p*	0.000 *	0.000 *	0.813 *	0.000 *	0.703	0.632	0.750
	η^2^p	0.072	0.121	0.001	0.021	0.001	0.002	0.001
L12	F	140.228	20.834	8.694	0.012	0.956	0.347	0.070
	*p*	0.000 *	0.000 *	0.000 *	0.912	0.385	0.707	0.932
	η^2^p	0.192	0.034	0.029	0.000	0.003	0.001	0.000
L13	F	165.111	144.513	4.281	16.621	0.091	0.558	0.056
	*p*	0.000 *	0.000 *	0.014 *	0.000 *	0.913	0.572	0.945
	η^2^p	0.219	0.197	0.014	0.027	0.000	0.002	0.000
L14	F	296.704	264.888	1.619	68.306	0.172	0.296	0.337
	*p*	0.000 *	0.000 *	0.199	0.000 *	0.842	0.744	0.714
	η^2^p	0.335	0.310	0.005	0.104	0.001	0.001	0.001
L15	F	83.502	43.353	10.621	7.122	1.110	1.133	3.357
	*p*	0.000 *	0.000 *	0.000 *	0.008 *	0.330	0.323	0.036 *
	η^2^p	0.124	0.069	0.035	0.012	0.004	0.004	0.011
L16	F	8.758	0.333	20.991	0.005	1.572	0.636	0.507
	*p*	0.003 *	0.564	0.000 *	0.945	0.208	0.530	0.602
	η^2^p	0.015	0.001	0.067	0.000	0.005	0.002	0.002
TT	F	555.605	101.015	8.254	0.011	0.144	0.183	0.116
	*p*	0.000 *	0.000 *	0.000 *	0.915	0.865	0.833	0.891
	η^2^p	0.485	0.146	0.027	0.000	0.000	0.001	0.000

Note: * marked values were significant when *p* ≤ 0.05; L-lap; TT-total time spent from the start until the end; G-gender; CS-competitive stage (semi-final/final); SL-skaters’ level; G × CS-interaction effect between gender and competitive stage; G × SL-interaction effect between gender and skaters’ level; CS × SL-interaction effect between competitive stage and skaters’ level; G × CS × SL-interaction effect among gender, competitive stage, and skaters’ level.

## Data Availability

Data is available by contacting the corresponding authors.

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
