# Peer review of "Analysis of Pacing Behaviors on Mass Start Speed Skating"

_ijerph, 2022, doi:10.3390/ijerph191710830_

Round 1

Reviewer 1 Report

Thank you for submitting your paper for review. My comments are as follows:

-You need to make a stronger case for the importance of the topic. That is, why should readers of the journal be interested in pacing behaviours of speed skaters? You begin to consider this when you discuss the potential implications of the findings for coaches. This could be developed further. For example, in the final sentence of the paper, you indicate that the findings could help coaching staff.  If you are able to elaborate on this more fully (ie how might they help specifically), that would add importantly to the contribution of the paper.

-In the final section, it would be helpful to more explicitly re-engage with existing literature. This will allow you to clearly note how your paper is contributing to and extending existing knowledge.

-Table 2 is very detailed. It would be helpful to explain more fully what this table is showing. Don't assume its meaning is self explanatory. You need to undertake some more interpretation of the Table to guide the reader.

-Line 210 mentions previously stated hypotheses. Would be good to restate these here, and consider in more detail.

-Around line 310, you discuss differences in definitions of skill level and how they change. This needs more explanation as to why the definitions are changing at this point.

-In the opening, you mention the forthcoming 2022 Winter Olympics. Given these have now been held, you can change the tense.

-The paper needs a careful proof read, in terms of language, grammar and written expression.

Author Response

Dear reviewer,

Thank you for reading our manuscript so carefully and giving us very detailed comments. It was very helpful for our study. We revised all the incorrect points in our manuscript according to your recommendations, and the revision is provided point by point as follows: 

Point 1: You need to make a stronger case for the importance of the topic. That is, why should readers of the journal be interested in pacing behaviors of speed skaters? You begin to consider this when you discuss the potential implications of the findings for coaches. This could be developed further. For example, in the final sentence of the paper, you indicate that the findings could help coaching staff.  If you are able to elaborate on this more fully (ie how might they help specifically), that would add importantly to the contribution of the paper.

Response 1: Line 44-53: we rephrase the expression to a stronger case for the importance of the topic. Also, Line 372-376, we elaborate on the contribution of the paper more specifically (how to help coaches and skaters to make decisions before and during the race).

Point 2: In the final section, it would be helpful to more explicitly re-engage with existing literature. This will allow you to clearly note how your paper is contributing to and extending existing knowledge.

Response 2: Line 352-354: we re-engaged with existing literature in the final section.

Point 3: Table 2 is very detailed. It would be helpful to explain more fully what this table is showing. Don't assume its meaning is self explanatory. You need to undertake some more interpretation of the Table to guide the reader.

Response 3: Line 132-135: we make more interpretation of the table 2-the meaning of TT, G×CS, G×SL, CS×SL, and G×CS×SL.

 Point 4: -Line 210 mentions previously stated hypotheses. Would be good to restate these here, and consider in more detail.

Response 4: Line 218-223: we restate the hypotheses here, and it is now easier to see the difference between the final results and previous hypotheses.

Point 5: Around line 310, you discuss differences in definitions of skill level and how they change. This needs more explanation as to why the definitions are changing at this point.

Response 5: Line 298-300: we reworded the expression in definitions of skill level. Also, it is explained specifically in line 331-333.

 Point 6: In the opening, you mention the forthcoming 2022 Winter Olympics. Given these have now been held, you can change the tense.

Response 6: Line 35-36: Since the 2022 Winter Olympics has been held, we deleted this sentence: it will be an official event for the 2022 Beijing Winter Olympic Games.

Point 7: The paper needs a careful proof read, in terms of language, grammar and written expression.

Response 7: we polished the English language, grammar and written expression through the manuscript. Thanks for your comments one more time.

Best regards

Reviewer 2 Report

I’d like to thank the Authors for the opportunity to review the article submitted to the International Journal of Environmental Research and Public Health.

Thank you for your paper, the paper is a timely and important issue to explore that I believe will merit publication.

This is the best paper I have ever reviewed!

Congratulations to all authors!

The manuscript present a compelling set of data. It is very well-written,  clearly and all of the conclusions are sound.

 Is the title not too long? In my opinion “Analysis of Pacing Behaviors on Mass Start Speed Skating’ is enough J

The originality or added value of this study should be included in the abstract.

In addition, in the abstract: The originality or added value of this study should be included. Besides, in my opinion, in the sentence ‘The one-way ANOVA for re-18 peated measures was run to compare the pacing behavior on each lap, and the three-way ANOVA 19 for repeated measures was used to identify the influence of SL, CS, and G on pacing behaviors and 20 total time spent’ , ‘run’ word should be replaced with the word ‘use’. I think that  it would be more appropriate.

 Best regards. 

Author Response

Dear reviewer,

Thank you for reading our manuscript so carefully and giving us very detailed comments. It was very helpful for our study. We revised all the incorrect points in our manuscript according to your recommendations, and the revision is provided point by point as follows: 

Point 1: Is the title not too long? In my opinion “Analysis of Pacing Behaviors on Mass Start Speed Skating’ is enough

Response 1: we deleted “According to Skater’s Level, Competition Stage and Gender”, now the title is “Analysis of Pacing Behaviors on Mass Start Speed Skating”.

Point 2: The originality or added value of this study should be included in the abstract.

Response 2: Line 14-15: we added originality to the abstract.

Point 3: Besides, in my opinion, in the sentence ‘The one-way ANOVA for re-peated measures was run to compare the pacing behavior on each lap, and the three-way ANOVA 19 for repeated measures was used to identify the influence of SL, CS, and G on pacing behaviors and total time spent’, ‘run’ word should be replaced with the word ‘use’. I think that it would be more appropriate.

Response 3: Line 20: we replace “run” with “used”

Besides, we also polished the English language through the manuscript. Thanks for your comments one more time.

Best regards

Reviewer 3 Report

Dear authors,

Thank you for the opportunity provided by you to study such an interesting and new article.

I will bring you below some of my opinions regarding what I would like to be clearer to increase the quality of this article:

1. this source cannot be identified, accessed: LI ZW, Yang J, Li HX. Research on women's collective starting tactics in speed skating. China Winter Sports. 2020;42(3):12-372 6.

2. this source cannot be identified, accessed: Wang XD, Hou YL, XuW.Q. Research on kinematics characteristics of 1000 M short track speed skating. China Sports Science 367 And Technology. 2018;54(5):127-31.

3. The research does not have the ethics approval of the research center, institute or university. Or at least I can't identify it in the article.

4. L140 - 3.3. Influence of CL, SL, and G on pacing behavior – instead of CL I think it should be CS????

5. L147-150, L153-164, L179-196 – presentation style too rigid. It's the statistics, ok, but it could be described more simply and with a minimum of explanation or argument, for readers who would like to understand more clearly

6. L220-221 , L225-226 ~~due to the inertial factor ……~~~ - please be more explicit what do you mean when you say inertial factor? Develop, because it was not a variable taken into account, and this factor could influence the behavior of athletes, regardless of level and gender

7. L278- There was an interaction for G and CL …..??? instead of CL I think it should be CS????

8. In the discussion chapter (L251-288) many phrases already written in the statistics subsections are repeated... 3.2. and 3.3. .... please develop and reformulate. The discussions should clarify the data generated by the statistical analysis, make them easier to understand. Here, with few exceptions, they are repeated without making any contribution to the reader's understanding of the data

Author Response

Dear reviewer,

Thank you for reading our manuscript so carefully and giving us very detailed comments. It was very helpful for our study. We revised all the incorrect points in our manuscript according to your recommendations, and the revision is provided point by point as follows:

Point 1: this source cannot be identified, accessed: LI ZW, Yang J, Li HX. Research on women's collective starting tactics in speed skating. China Winter Sports. 2020;42(3):12-372 6.

Response 1: Line 406: This source is obtained from the local journal in China. And we have cited a new source from the international journal, accessed: Konings, M.J.; Hettinga, F. J. Objectifying tactics: athlete and race variability in elite short-track speed skating. International journal of sports physiology and performance 2018, 13(2): 170-5

Point 2: this source cannot be identified, accessed: Wang XD, Hou YL, XuW.Q. Research on kinematics characteristics of 1000 M short track speed skating. China Sports Science and Technology. 2018;54(5):127-31.

Response 2: Line 412: This source is obtained from the local journal in China. And we have cited a new source from the international journal, accessed: Jin, W.; Yanhong, S. Investigating Tactics Characteristics of Mass-Start Event of Speed Skating in Pyeongchang Winter Olympics. Revista de Psicología del Deporte (Journal of Sport Psychology) 2021, 30, (2), 263-272

Point 3: The research does not have the ethics approval of the research center, institute or university. Or at least I can't identify it in the article.

Response 3: Line 385-386: we have added the information on the ethics approval of the university.

Point 4: L140 - 3.3. Influence of CL, SL, and G on pacing behavior – instead of CL I think it should be CS????

Response 4: We have corrected “CL” to “CS”. Also, we checked the full text.

Point 5: L147-150, L153-164, L179-196 – presentation style too rigid. It's the statistics, ok, but it could be described more simply and with a minimum of explanation or argument, for readers who would like to understand more clearly

Response 5: Line 157-168, Line 172-178, and Line 182-203: we simplified the explanation of the results.

Point 6: L220-221, L225-226 ~~due to the inertial factor ……~~~ - please be more explicit what do you mean when you say inertial factor? Develop, because it was not a variable taken into account, and this factor could influence the behavior of athletes, regardless of level and gender

Response 6: Line 233-236, line 241-242:  we tried to explain the inertial factor more explicitly. Inertial factor in the text means that skaters sprint with very high speed on laps 4, 8, and 12 (point laps), so skaters do not need to sprint with high speed on laps 5, 9, and 13 but they do get fast since they can obtain extra speed from lap 4, 8, and 12. Skaters normally adopt this strategy during the race.

Point 7: L278- There was an interaction for G and CL …..??? instead of CL I think it should be CS????

Response 7: We have corrected “CL” to “CS”. Also, we checked the full text.

Point 8: In the discussion chapter (L251-288) many phrases already written in the statistics subsections are repeated... 3.2. and 3.3. .... please develop and reformulate. The discussions should clarify the data generated by the statistical analysis, make them easier to understand. Here, with few exceptions, they are repeated without making any contribution to the reader's understanding of the data.

Response 8: Line 267-308:: We reformulated and developed the discussion to make the data clear and readable.

Besides, we also polished the English language through the manuscript. Thanks for your comments one more time.

Best regards

Round 2

Reviewer 1 Report

Thank you for addressing the comments appropriately.

My only recommendation now is that there is a need for a final very careful proof reading, to improve the written expression.

Author Response

Dear reviewer,

Thank you for your careful evaluation and helpful comments on our manuscript. We have carefully done the final proof reading and hope that the manuscript is clearer and more compelling.

Best regards